

# Characterization of *Streptomyces* sp. KB1 and its cultural optimization for bioactive compounds production

Monthon Lertcanawanichakul[1,2] and Tuanhawanti Sahabuddeen[3]

[1] School of Allied Health Sciences, Walailak University, Thasala, Thaiburi, Nakhon Si Thammarat, Thailand
[2] Food Technology and Innovation Research Center of Excellence, Walailak University, Thaiburi, Thasala, Nakhon Si Thammarat, Thailand
[3] Research Unit of Natural Product Utilization, Walialk University, Thaiburi, Thasala, Nakhon Si Thammarat, Thailand

## ABSTRACT

**Background**. Bioactive compounds (BCs) from natural resources have been extensively studied because of their use as models in the development of novel and important medical and biopreservative agents. One important source of BCs is microorganisms, particularly terrestrial bacteria of the order Actinomycetales.

**Methods**. We characterized *Streptomyces* sp. KB1 by observing its morphology, physiology, and growth on different media using biochemical tests, optimizing cultural conditions by changing one independent variable at a time.

**Results**. *Streptomyces* sp. KB1 (TISTR 2304) is a gram-positive and long filamentous bacteria that forms straight to flexuous (rectiflexibile) chains of globose-shaped and smooth-surfaced spores. It can grow under aerobic condition s only at a temperature range of 25–37 °C and initial pH range of 5–10 in the presence of sodium chloride 4% (w/v). Therefore, it is considered an obligate aerobe, mesophilic, neutralophilic, and moderately halophilic bacteria. The isolate grew well on peptone-yeast extract iron, Luria Bertani (LB), and a half-formula of LB (LB/2), but could not grow on MacConkey agar. It utilized fructose, mannose, glucose, and lactose as its carbon source along with acid production and showed positive reactions to casein hydrolysis, gelatin liquefaction, nitrate reduction, urease, and catalase production. *Streptomyces* sp. KB1 (TISTR 2304) could produce the maximum number of BCs when 1% of its starter was cultivated in a 1,000 ml baffled flask containing 200 ml of LB/2 broth with its initial pH adjusted to 7 with no supplemental carbon source, nitrogen source, NaCl, or trace element at 30 °C, shaken at 200 rpm in an incubator for 4 days.

## INTRODUCTION

Microorganisms have been used in natural products and processes that benefit and improve human socioeconomic lifestyles since the days of early civilization. Actinomycetes are a group of microorganisms that are extensively used in natural and manmade environments. They play an important role in producing secondary metabolites of novel structures, including antibacterial, antifungal, antitumor, antiprotozoic, and antiviral properties,

Corresponding author
Monthon Lertcanawanichakul,
Lmonthon55@gmail.com

as well as vitamins and enzymes (*Priya et al., 2012*). Currently, about 75% of known antibiotics are isolated from actinomycetes, and of these, two-thirds are produced from *Streptomyces* (*Mohanraj & Sekar, 2013*). The genus *Streptomyces* was first discovered by *Waksman & Henrici (1943)*. They are gram-positive and obligate aerobe bacteria that resemble fungi in their branching filamentous structure and grow in various environments (*Nayaka & Babu, 2014*). Numerous classifications have been devised to accommodate the growing number of *Streptomyces* species (*Laidi et al., 2006*). *Streptomyces* are an obligate, aerobe bacteria that are chemoorganotrophic, non-fastidious, filamentous, spore forming, and non-motile with high guanine-cytosine (G-C) content (approximately 69–78%) in their DNA (*Anderson & Wellington, 2001*; *Nayaka & Babu, 2014*). They are free-living, saprophytic bacteria that are widely distributed in soil, water, and colonizing plants (*Rahman et al., 2010*). The original identification of *Streptomyces* was performed following morphological observations. Although morphology is still an important characteristic to consider for the description of taxa, it is not adequate when used alone to differentiate between many genera. Chemotaxonomic criteria have provided taxonomists with a set of reliable and reproducible tools for identifying the genus *Streptomyces*. In addition, biochemical, cultural, and physiological characteristics have been also used to identify at the species level. However, it has been suggested that this approach depends on the survey of extensive numerical taxonomy of *Williams, Goodfellow & Alderson (1989)*. The 16s rDNA sequence analysis has proved to be a very important tool for *Streptomyces* systematic analysis, as well as helpful in identifying newly isolated strains of *Streptomyces* (*Kim et al., 1999*). There is increasing interest in the isolation of a novel *Streptomyces* species as they are very potent producers of bioactive compounds (BCs). In our previous study, the producing strain of *Streptomyces* isolated from air samples from Ao-nang, Krabi Province, Thailand, showed broad-spectrum antimicrobial activity against a representative of yeasts and gram-positive and gram-negative bacteria, including methicillin-resistant *Staphylococcus aureus* (MRSA). It was identified using 16S rDNA sequencing, named *Streptomyces* sp. KB1, and deposited in the GenBank database and the Thailand Institute of Scientific and Technological Research under the accession numbers KF939581.1 and TISTR 2304, respectively (*Chawawisit et al., 2015*).

In inappropriate environments, most *Streptomyces* usually produce secondary metabolites that kill or inhibit the growth of related microorganisms (*Hasani, Kariminik & Issazadeh, 2014*). Therefore, most secondary metabolites are generally secreted outside the cell at the late growth phase. Secondary metabolites that have biological, anti-bacterial, anti-fungal, anti-viral, and anti-malarial properties are called bioactive compounds (BCs). These active molecules are chemically and taxonomically diverse compounds that are not essential for an organism's normal growth, development, or reproduction (*Donadio et al., 2002*). The temporal nature of BCs to synthesize is controlled by genetic materials that are greatly connected with primary metabolism and influenced by environmental manipulation. The biosynthesis of BCs is often activated by the exhaustion of a nutrient, addition of an inducer, or decrease of growth rate and different environmental signals produced by other microorganisms in the environment (*Bibb, 2005*; *Sharma & Thakur, 2020*). Waksman and colleagues (*1961*) reported that the ability of *Streptomyces* cultures to

form BCs was not a fixed property, but one that could be greatly increased or completely lost under different conditions of nutrition and cultivation. However, minor changes in the nature and type of carbon, nitrogen, phosphate sources, and trace elements have also affected BCs' biosynthesis of *Streptomyces* (*Abbanat, Maiese & Greenstein, 1999*). Moreover, the biosynthesis of BCs tends to decrease when metal-ion-deficient media are used and the inocula are incubated for long periods and at high temperatures (*Reddy, Ramakrishna & Rajagopal, 2011*). In addition to nutrients, media may also contain various inhibitors of microbial growth that may affect BC biosynthesis (*Liu et al., 2013*). Therefore, designing an appropriate culture medium is very important as medium composition can significantly affect the yield of BCs. Most BCs are generally extra-cellular and their isolation in high purity from complex fermentation broth requires the application of a combination of various techniques such as solvent extraction, chemical precipitation, chromatography, and HPLC purification (*Mehdi et al., 2006*; *Sharma & Thakur, 2020*).

This study was performed to characterize the micro-morphology, chemotaxonomy, growth on different media, physiology, and biochemistry of *Streptomyces* sp. KB1, study the effect of various cultural conditions on the production of BCs, and determine the optimal cultural conditions for maximum production. Our preliminary findings implied that BCs from *Streptomyces* sp. KB1 are important and efficient compounds that may be used as lead compounds in the discovery of new anti-MRSA drugs in the future.

## MATERIALS AND METHODS

### Microorganisms, media, and cultural conditions

#### Producing strain

*Streptomyces* sp. KB1 was cultured on a half-formula of Luria Bertani (LB/2) (Himedia, India) agar medium (5 g/l Tryptone, 2.5 g/l yeast extract, 5 g/l NaCl and 15 g/l agar powder) at 30 °C in a static incubator (WTB Binder, Tuttlingen, Germany) for 5 days. A single colony was inoculated into $25 \times 150$ mm of the screw cap test tube containing 10 ml of LB/2 broth medium, incubated at 30 °C, 200 revolutions per minute (rpm) in a shaking incubator (Thermo Scientific, Waltham, MA, USA) for 2 days. The cells were stored in 15% glycerol at $-80$ °C until use.

#### Indicator strains

*Staphylococcus aureus* TISTR 517, obtained from the Thailand Institute of Scientific and Technological Research, Pathum Thani Province, Thailand, was cultured in Luria Bertani (LB) agar medium (Himedia, India) at 37 °C in a static incubator for 24 h. A single colony was inoculated into $25 \times 150$ mm of the screw cap test tube containing 10 ml of LB broth medium, incubated at 37 °C, 200 rpm in a shaking incubator for 24 h. The cells were stored in 15% glycerol at $-80$ °C until use.

#### Starter preparation

We inoculated $1 \times 1$ cm$^2$ of initial streak of 5-day culture of *Streptomyces* sp. KB1 into $25 \times 150$ mm of screw cap test tube containing 10 ml of LB/2 broth medium. The inoculum was incubated at 30 °C, 200 rpm in a shaking incubator for 2 days, and assigned as the starter.

## Morphological characterization

A loopful of *Streptomyces* sp. KB1 starter was streaked on yeast extract-malt extract (YE-ME) agar medium and incubated at 30 °C for 7 days. A plug of the culture was removed and fixed in glutaraldehyde (2.5% v/v), washed with water, and post-fixed in osmiumtetroxide (1% w/v) for 1 h. The sample was washed twice with water and dehydrated in ascending ethanol before drying in acritical point drying apparatus (PolaronE3000) and finally coated with gold palladium and observed by scanning electron microscope (SEM) (Quanta 400; FEI, Hillsboro, Oregon, USA).

## Chemotaxonomic analysis

Cells used for chemotaxonomic analysis were obtained after incubating *Streptomyces* sp. KB1 at 30 °C for 2 days in a half-formula of Luria Bertani broth (pH 7.0). Isomers of diaminopimelic acid in the whole-cell hydrolysate were determined using thin-layer chromatography according to the method of *Schön & Groth (2006)*. Whole-cell sugars were analyzed according to the method of *Becker, Lechevalier & Lechevalier (1965)*.

## Cultural characterization

Ten µl of starter of *Streptomyces* sp. KB1 was streaked on YE-ME agar, oatmeal agar, inorganic salt-starch agar, glycerol-asparagine agar, peptone-yeast extract iron agar, tyrosine agar, LB agar, LB/2 agar, and nutrient agar according to the methods of *Shirling & Gottlieb (1966)*. After 14 days of incubation at 30 °C, the growth, texture, substrate mycelia, aerial mycelia, spore formation, color, and soluble pigment on various media were observed.

## Biochemical and physiological characterization

The biochemical and physiological characteristics were performed according to the methods of *Shirling & Gottlieb (1966)*. The biochemical characteristics included growth on MacConkey agar, indole test, methyl red test, voges proskauer test, citrate utilization, casein hydrolysis, urea hydrolysis, gelatin hydrolysis, nitrate reduction, $H_2S$ production, cytochrome oxidase test, and catalase test. The different sugars viz. arabinose, ribose, rhamnose, mannitol, fructose, sucrose, galactose, mannose, inositol, glucose, glycerol, maltose, and lactose were used to determine the acid production of *Streptomyces* sp. KB1. The physiological characterization of *Streptomyces* sp. KB1 was carried out by analyzing the growth at different temperatures ranging from 20 to 42 °C, different pH ranging from 4 to 12, and growth under anaerobic conditions. NaCl tolerance was determined by cultivating *Streptomyces* sp. KB1 on LB/2 agar medium supplemented with 0, 1, 2, 3, 4, 5, 6, 7, 8, 9, and 10% (w/v) NaCl. The maximum NaCl concentration in the medium that allowed any growth was recorded.

## Cultural optimization for BC production

The effect of various cultural conditions (both nutritional and physiological) on the production of BCs by *Streptomyces* sp. KB1 was determined. The parameters that were studied included the formula of LB medium, inoculum, incubation period, incubation temperature, size of baffled flask, agitation, initial pH, carbon source, nitrogen source, salt concentration, starch concentration, and trace elements.

Table 1  **Antimicrobial activity.** The anti *S. aureus* activity of Streptomyces across competitive factors.

| Formula | Ingredient of each formula | | | Named as |
|---|---|---|---|---|
| | Tryptone (g/L) | Yeast extract (g/L) | NaCl (g/L) | |
| Full-formula | 10.00 | 5.00 | 10.00 | LB |
| Reduce-formula: | | | | |
| 1 | 5.00 | 2.50 | 5.00 | LB/2 |
| 2 | 3.33 | 1.67 | 3.33 | LB/3 |
| 3 | 2.50 | 1.25 | 2.50 | LB/4 |
| 4 | 2.00 | 1.00 | 2.00 | LB/5 |
| 5 | 1.67 | 0.83 | 1.67 | LB/6 |

### Selection of basal production medium

To determine the optimal cultural conditions for *Streptomyces* sp. KB1's maximum BC production, LB medium was selected as the basal production medium.

### Effect of LB medium formula

LB medium was prepared as full-formula and applied as a reduced formula viz. LB/2, LB/3, LB/4, LB/5, and LB/6 broth as illustrated in Table 1. We inoculated 1% starter of *Streptomyces* sp. KB1 (TISTR 2304) into a 1,000 ml baffled flask containing 200 ml of each formula LB broth medium. All inoculum were incubated at 30 °C at 200 rpm in a shaking incubator for 14 days. The culture broth was harvested and the cell sediment was separated using the centrifugation technique. The supernatant's anti-*S. aureus* TISTR 517 activity was tested using the agar well diffusion method (*Kekuda et al., 2012*) in order to determine the best LB medium formula for the production of BCs.

### Effect of inoculum concentration

Varied starter concentrations (0.5, 1, 2, 3, 4, and 5%) were inoculated into a 1,000 ml baffled flask containing 200 ml of LB/2 broth medium. Each inoculum was incubated at 30 °C at 200 rpm in a shaking incubator for 7 days. The culture broth was harvested and the cell sediment was separated using the centrifugation technique. The supernatant's anti-*S. aureus* TISTR 517 activity was measured using the agar well diffusion method in order to determine the optimal inoculum concentration for production of *Streptomyces* sp. KB1 (TISTR 2304) BCs.

### Effect of incubation period, incubation temperature, and size of baffled flask

We inoculated 1% starter of *Streptomyces* sp. KB1 (TISTR 2304) into 250, 500, and 1,000 ml baffled flasks containing 50, 100, and 200 ml of LB/2 broth medium, respectively. All inoculum were cultivated at 25, 30, and 35 °C, 200 rpm in a shaking incubator for 14 days. On days 1, 2, 3, 4, 5, 6, 7, 8, 9, 10, 11, 12, 13, and 14, the culture broth was harvested and the cell sediment was separated using the centrifugation technique. The supernatant's anti-*S. aureus* TISTR 517 activity was tested using the agar well diffusion method to evaluate the optimal incubation period, incubation temperature, and size of baffled flask for production of BCs.

### Effect of agitation

We inoculated 1% starter of *Streptomyces* sp. KB1 (TISTR 2304) into four 1,000 ml baffled flasks containing 200 ml of LB/2 broth medium. Each flask was incubated at 30 °C and 0, 100, 150, 200, and 250 rpm in a shaking incubator for 4 days. After incubation, the culture broth was harvested and the cell sediment was separated using the centrifugation technique. The supernatant anti-*S. aureus* TISTR 517 activity was tested using the agar well diffusion method to evaluate the optimal agitation for production of BCs.

### Effect of initial pH of medium

We inoculated 1% starter of *Streptomyces* sp. KB1 (TISTR 2304) into a 1,000 ml baffled flask containing 200 ml of LB/2 broth medium with adjusted initial pH of 5, 6, 7, 8, and 9. All inoculum were incubated at 30 °C, 200 rpm in a shaking incubator for 4 days. The culture broth was harvested and the cell sediment was separated using the centrifugation technique. The supernatant anti-*S. aureus* TISTR 517 activity was tested using the agar well diffusion method to determine the optimal initial pH of medium for maximal production of BCs.

### Effect of carbon sources

We inoculated 1% starter of *Streptomyces* sp. KB1 (TISTR 2304) into a 1,000 ml baffled flask containing 200 ml of LB/2 broth medium supplement with 1% different carbon sources viz. glucose, maltose, sucrose, glycerol, and starch. All inocula were incubated at 30 °C at 200 rpm in a shaking incubator for 4 days. The culture broth was harvested and the cell sediment was separated using the centrifugation technique. The supernatant anti-*S. aureus* TISTR 517 activity was tested using the agar well diffusion method to evaluate the optimal carbon source for production of BCs.

### Effect of nitrogen sources

We inoculated 1% starter of *Streptomyces* sp. KB1 (TISTR 2304) into 1,000 ml baffled flask containing 200 ml of LB/2 broth medium supplement with 0.3% different nitrogen sources viz. casein, peptone, beef extract, malt extract, urea, and ammonium sulphate. All inocula were incubated at 30 °C at 200 rpm in a shaking incubator for 4 days. The culture broth was harvested and the cell sediment was separated using the centrifugation technique. The supernatant anti-*S. aureus* TISTR 517 activity was tested using the agar well diffusion method to evaluate the optimal nitrogen source for production of BCs.

### Effect of salt concentration

We inoculated 1% starter of *Streptomyces* sp. KB1 (TISTR 2304) into a 1,000 ml baffled flask containing 200 ml of LB/2 broth medium supplement with NaCl at different concentrations viz. 0.0, 0.5, 1.0, 1.5, 2.0, 2.5, 3.0, 3.5, and 4.0% (w/v) and incubated at 30 °C at 200 rpm in a shaking incubator for 4 days. The culture broth was harvested and the cell sediment was separated using the centrifugation technique. The supernatant anti-*S. aureus* TISTR 517 activity was tested using the agar well diffusion method to evaluate the optimal salt concentration for production of BCs.

*Effect of trace elements*

We inoculated 1% starter of *Streptomyces* sp. KB1 (TISTR 2304) into a 1,000 ml baffled flask containing 200 ml of LB/2 broth medium supplemented with 0.1% of trace elements viz. $FeSO_4$, $MgSO_4$, $CaCO_3$, $K_2HPO_4$, $KH_2PO_4$, KCl, and $KNO_3$ and incubated at 30 °C at 200 rpm in a shaking incubator for 4 days. The culture broth was harvested and the cell sediment was separated using the centrifugation technique. The supernatant anti-*S. aureus* TISTR 517 activity was tested using the agar well diffusion method to evaluate the optimal trace elements for production of BCs.

## Anti- *S. aureus* TISTR 517 activity assay

Anti-*S.aureus* TISTR 517 activity of BCs was investigated using the agar well diffusion method (*Kekuda et al., 2012*). A single colony of 24 h-culture of *S. aureus* TISTR 517 was inoculated into $15 \times 150$ mm of screw cap test tube containing 5 ml of LB broth medium and incubated at 37 °C, 200 rpm in a shaking incubator for 24 h. Cell suspension was approximately adjusted to a 0.5 McFarland standard turbidity using 0.85% NaCl and swabbed on sterilized MHA plates using a sterile cotton swab and drilled well with a five mm-diameter sterile cork borer. We transferred 50 µL of supernatant and medium that was assigned as the control to the labeled well. The plate was incubated at 37 °C in a static incubator for 24 h and observed the zone of inhibition formed around the well.

## Statistical analysis

All investigations were performed in triplicate. The obtained data were analyzed using SPSS software version 17. One-way analysis of variance (ANOVA) was employed and the level of significance was $P$ value <0.05. Post-hoc turkey analysis was used to investigate the differences across the data.

## RESULTS AND DISCUSSION

The strain KB1 was isolated from the air sample collected from Ao-nang, Krabi Province, Thailand. It was then identified using genetic methods and 16S rDNA sequence analysis and named *Streptomyces* sp. KB1 (*Chawawisit et al., 2015*). Analysis was performed according to the description in Bergey's manual of systematic bacteriology (*Williams, Goodfellow & Alderson, 1989*) and the methods for characterization of *Streptomyces* species (*Shirling & Gottlieb, 1966*).

## Morphological characteristics

The micro-morphological characteristics were observed after cultivating *Streptomyces* sp. KB1 on YE-ME agar medium at 30 °C for 14 days. *Streptomyces* sp. KB1 is a gram-positive and long filamentous bacteria that can produce aerial mycelia and spores. A scanning electron micrograph revealed that it formed straight to flexuous (rectiflexibile) chains of globose-shaped and smooth-surfaced spores. Each chain contained <15 spores (oligosporous) that developed on lateral branches of aerial hyphae (Fig. 1).

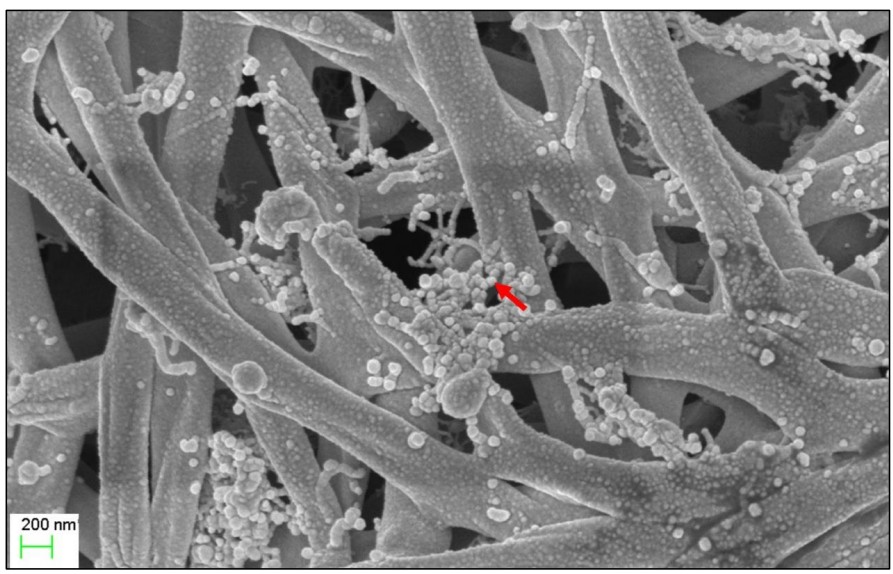

**Figure 1** Scanning electron microscope of *Streptomyces* KB1.

## Cultural characterization

The cultural characteristics were observed after cultivating *Streptomyces* sp. KB1 on different agar media at 30 °C in a static incubator for 14 days. We found that *Streptomyces* sp. KB1 showed good growth on peptone-yeast extract iron, LB, and LB/2 medium; moderate growth on YE-ME, oatmeal, and glycerol-asparagine medium; and poor growth on inorganic salt-starch, tyrosine, and nutrient medium (Fig. 2). It has both rough and smooth textures. The colors of substrate and aerial mycelium were different in each medium. *Streptomyces* sp. KB1 could form spores that were light-yellow and white color on YE-ME and glycerol-asparagine medium, respectively, but could not produce soluble pigments in all media (Table 2).

## Physiological characteristics

The physiological characteristics of *Streptomyces* sp. KB1 are shown in Table 3. We found that *Streptomyces* sp. KB1 could grow under aerobic conditions at a temperature range of 25–37 °C and initial pH range of 5–10 in the presence of <4% NaCl (w/v). Additionally, the optimal growth of *Streptomyces* sp. KB1 was observed at 30 °C and an initial medium pH of 7 in the presence of 1% (w/v) NaCl. These results indicated that *Streptomyces* sp. KB1 is an obligate aerobe, mesophilic, neutralophilic, and moderate halophilic bacteria.

## Biochemical characteristics

*Streptomyces* sp. KB1 could utilize fructose, mannose, glucose, and lactose as its carbon source along with acid production. Arabinose, ribose, rhamnose, mannitol, sucrose, galactose, inositol, glycerol, and maltose were not utilized by this strain and showed no growth on MacConkey agar. The biochemical tests for casein hydrolysis, gelatin liquefaction, nitrate reduction, urease and catalase production showed positive results, but

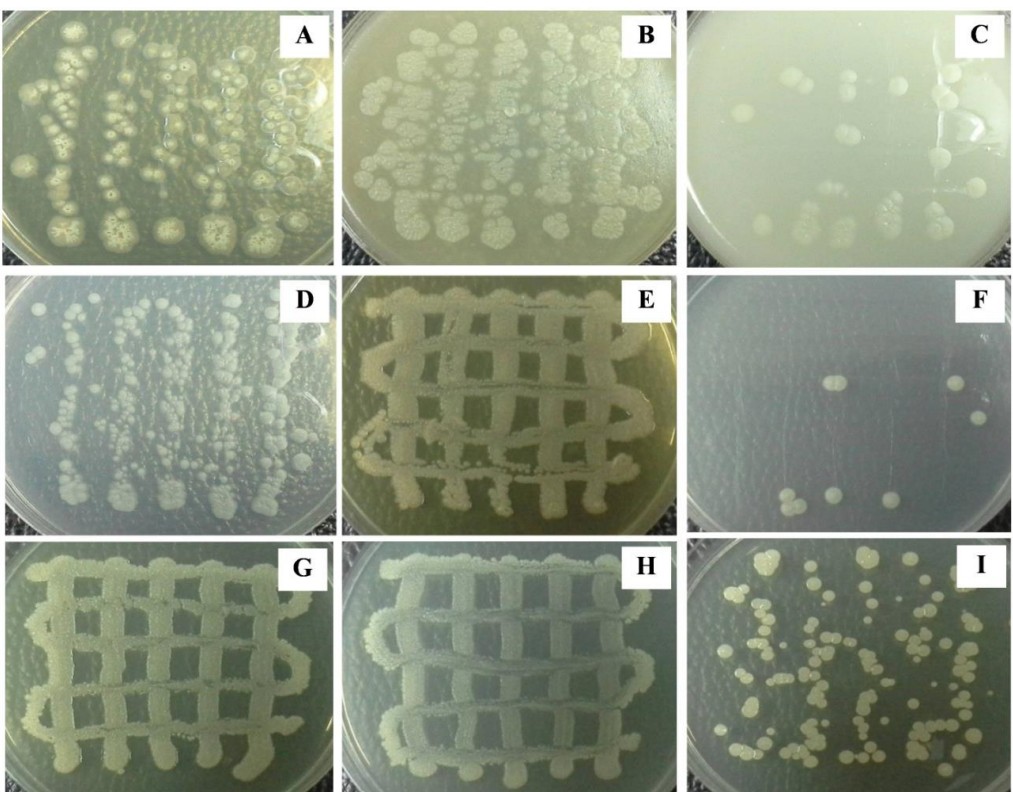

**Figure 2** (A–I) Colony morphology of *Streptomyces* in various types of media.

starch hydrolysis, citrate utilization, H$_2$S production, indole production, methyl red test, and voges proskauer test were negative (Table 4).

## Cultural optimization of *Streptomyces* sp. KB1 for BC production

The ability of *Streptomyces* cultures to form BCs was not a fixed property but could be greatly increased or completely lost under different nutrition and cultivation conditions (*Waksman, 1961*). Using preliminary tests and the agar well diffusion method, we found that BCs showed the most anti-*S. aureus* activity observed from the zone of inhibition (Fig. 3). In order to determine the maximum production of BCs of *Streptomyces* sp. KB1 (TISTR 2304), cultural optimization was essential. The effects of LB medium formula, carbon sources, nitrogen sources, salt concentration, trace elements, and other physical parameters on BC production of *Streptomyces* sp. KB1 were optimized by changing one independent variable at a time. Previous studies usually started the optimization process by selecting the basal production medium. We found that the complex media, especially yeast extract glucose broth medium and yeast extract-malt extract-dextrose broth medium, were often used as basal production medium (*Sultan et al., 2002*; *Narayana & Vijayalakshmi, 2008*; *Sharon, Daniel & Shenbagarathai, 2014*). These media had both carbon and nitrogen sources that supported the production of *Streptomyces* sp. BCs. However, *Streptomyces* sp. KB1 could not produce BCs when it was cultivated in yeast extract-malt extract-dextrose

**Table 2 Growth in various.** Colony morphology in various media.

| Media | Property relevants | | | | | |
|---|---|---|---|---|---|---|
| | Growth | Texture | Substrate mycelia | Aerial mycelia | Spore formation and color | Soluble pigment |
| Yeast extract-Malt extract (ISP2) | Moderate | Rough | Pale-yellow | Abundant, White | Moderate, Light-yellow | Nil |
| Oatmeal (ISP3) | Moderate | Rough | White | Poor, White | Nil | Nil |
| Inorganic salts-starch (ISP4) | Poor | Smooth | White | Poor, White | Nil | Nil |
| Glycerol-asparagine (ISP5) | Moderate | Rough | White | Moderate, White | Poor, White | Nil |
| Peptone-yeast extract iron (ISP6) | Well | Rough | Pale-yellow | Abundant, Pale-yellow | Nil | Nil |
| Tyrosine (ISP7) | Poor | Rough | White | Poor, White | Nil | Nil |
| Luria Bertani | Well | Rough | Pale-yellow | Abundant, Pale-yellow | Nil | Nil |
| Half-formula of Luria Bertani | Well | Rough | White | Poor, White | Nil | Nil |
| Nutrient | Poor | Smooth | Pale-yellow | Poor, Pale-yellow | Nil | Nil |

**Notes.**
Nil, not produced.

broth medium. This phenomenon was also found by *Dekleva & Strohl (1987)* who observed that the production of Daunorubicin of *Streptomyces peucetius* was negatively affected by glucose. They suggested that this inhibitory effect is caused by carbon catabolite repression (CCR) (*Dekleva & Strohl, 1987*). CCR is the regulation of the secondary metabolism of microorganisms by carbon sources in the medium that repress the expression of certain genes and operons involved with secondary metabolites or BC biosynthesis (*Bruckner & Titgemeyer, 2002*). As a result, *Streptomyces* sp. KB1 (TISTR 2304) was used to attempt to cultivate in a complex medium, LB medium, which has a nitrogen source only. Surprisingly, *Streptomyces* sp. KB1 could produce BCs observed from the zone of inhibition against *Staphylococcus aureus* TISTR 517. However, other studies have reported that nutritional excess is directed towards the generation of cell mass rather than the production of BCs, and when the depletion of key nutrients occurs, it shifts the cell cycle to the stationary phase and signals the transition from primary to secondary metabolism, where these BCs are produced (*Kirk, Rossa & Bushell, 2000*). Therefore, in order to investigate the correlation of nutritional excess with BC production of *Streptomyces* sp. KB1 (TISTR 2304) and also reduce the fermentation time and costs, the effect of LB medium formula on BC production was optimized as the first priority.

### Effect of LB medium formula on BC production

The effect of different LB medium formulas (LB, LB/2, LB/3, LB/4, LB/5, and LB/6) on BC production of *Streptomyces* sp. KB1 (TISTR 2304) was estimated using the basis of diameter of zone of inhibition (Table 5). The largest diameter of inhibition zone was most significant in the full- and half-formulas of LB medium (LB and LB/2, respectively). This result indicated that *Streptomyces* sp. KB1 (TISTR 2304) could produce the maximum number of BCs when it was cultured in LB and LB/2 medium. However, the maximum number of BCs were produced on the 4th and 7th day of cultivation of *Streptomyces* sp. KB1 (TISTR 2304) in LB/2 and LB medium, respectively. BCs or bioactive secondary metabolites are not vital to cell survival itself, but are more so for the entire microorganism's

**Table 3** **Physiological characteristics of *Streptomyces* sp. KB1.** The growth of KB1 in various condition.

| Characteristics | Condition |
| --- | --- |
| Temperature for growth (°C) | |
|     Range (°C) | 25–37 |
|     Optimum (°C) | 30 |
| NaCl tolerance (%) | <4 |
|     Optimum NaCl for growth (%) | 1 |
| pH for growth | |
|     Range | 5–10 |
|     Optimum | 7 |
| Growth under anaerobic condition | − |

Notes.
−, No Growth.

survival or when the cell is not operating under optimal conditions, *e.g.*, when a nutrient source is depleted. Most BCs are produced during the end of the exponential phase or near the stationary phase of growth (*Berdy, 2005*). This phenomenon could explain why the depletion or limitation of LB medium causes the shift of cell cycle to the stationary phase and signals the transition from primary to secondary metabolism (*Abd-Allah & El-Mehalawy, 2002*). Therefore, *Streptomyces* sp. KB1 (TISTR 2304) cultured in LB/2 medium produced the maximum number of BCs faster than those cultured in LB medium. Because nutritional excess is directed towards the generation of cell mass rather than the production of BCs, *Streptomyces* sp. KB1 (TISTR 2304) cultured in LB medium consequently produced the equal number of BCs as those cultured in LB/2 medium. However, when *Streptomyces* sp. KB1 (TISTR 2304) was cultured in LB medium that reduced the formula 3-fold (LB/3), the production of BCs significantly decreased. This finding could suggest that the production of *Streptomyces* sp. KB1 (TISTR 2304) BCs depends on the formula of LB medium. Moreover, excess LB medium was not essential for the production of BCs of *Streptomyces* sp. KB1 (TISTR 2304). Therefore, a further optimization process was carried out using LB/2 medium as the basal production medium.

### Effect of inoculum concentration on BC production

The effect of inoculum concentration (0.5, 1.0, 2.0, 3.0, 4.0, and 5.0%) of *Streptomyces* sp. KB1 (TISTR 2304) on the production of BCs was estimated using the basis of diameter of zone of inhibition presented in Table 5. It was revealed that *Streptomyces* sp. KB1 (TISTR 2304) produced the significantly lowest number of BCs when 0.5% inoculum of *Streptomyces* sp. KB1 (TISTR 2304) was cultured in LB/2 medium. However, when 1.0, 2.0, 3.0, 4.0, and 5.0% inoculum of *Streptomyces* sp. KB1 (TISTR 2304) was cultured in LB/2 medium, its ability to produce BCs was not significantly different. This finding indicated that an increase of inoculum concentration higher than 1.0% did not affect the production of BCs of *Streptomyces* sp. KB1 (TISTR 2304). Moreover, some studies have reported that the increase of inoculum concentration inhibited the production of BCs. It was discussed that the excess inoculum caused an increase in pH of the fermentation medium, leading to the suppression of BC production (*Hamdy et al., 2011*). *Nandhini & Selvam (2013)*

**Table 4** **Biochemical characteristics of *Streptomyces* sp. KB1.** Characteristics of KB1 in biochemical testing.

| Characteristics | Results |
|---|---|
| Gram staining | + |
| Growth on MacConkey agar | − |
| Starch hydrolysis | − |
| Casein hydrolysis | + |
| Citrate utilization | − |
| Gelatin liquefaction | + |
| $H_2S$ production | − |
| Indole production | − |
| Nitrate reduction | + |
| Methyl red test | − |
| Voges proskauer test | − |
| Triple sugar iron (TSI) | K/K − |
| Urease production | + |
| Catalase production | + |
| Diaminopimelic acid (DAP) | |
| Acid production with different sugars | |
|     Arabinose | − |
|     Ribose | − |
|     Rhamnose | − |
|     Mannitol | − |
|     Fructose | + |
|     Sucrose | − |
|     Galactose | − |
|     Mannose | + |
|     Inositol | − |
|     Glucose | + |
|     Glycerol | − |
|     Maltose | − |
|     Lactose | + |

**Notes.**
+, Growth; -, No Growth; K/K, Alkaline/Alkaline.

discussed how the increase of inoculum of *Streptomyces lavendulae* leads to the rapid use of nutrients for proliferation and biomass synthesis. However, when nutrients are depleted, there is a decrease of metabolic activity and production of BCs (*Nandhini & Selvam, 2013*). At the optimal inoculum concentration for BC production, there is a balance between the use of nutrients for proliferation, biomass synthesis, and the production of BCs. Therefore, our further optimization process was carried out using an inoculum concentration of 1.0%.

### Effect of incubation temperature, incubation period, and container volume
Incubation period, incubation temperature, and container volume played an important role in BC production and activity (*Bundale et al., 2015*). We inoculated 1% inoculum of

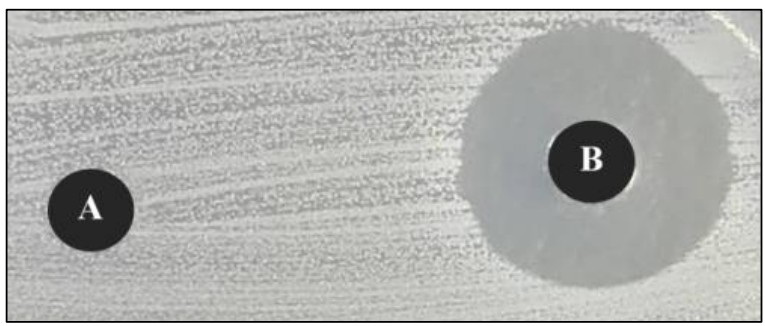

**Figure 3  Illustrated zone of inhibition formed around the well of medium broth (A), bioactive compounds (BCs) (B) against *Staphylococcus aureus.*** The zone of inhibition of BCs was greater than in the negative control (medium broth).

**Table 5  Effect of LB medium formula and inoculum concentration on antimicrobial compound production.** Each data point indicates the average antimicrobial activity across various conditions.

|  | Mean diameter of inhibition zone (mm)* |
| --- | --- |
| Formula |  |
| Full-formula: LB | 23.00 ± 0.00 |
| Reduce-formula: |  |
| LB/2 | 23.00 ± 0.00 |
| LB/3 | 20.67 ± 0.58 |
| LB/4 | 18.33 ± 0.58 |
| LB/5 | 15.67 ± 1.15 |
| LB/6 | 12.33 ± 0.58 |
| Inoculum concentration (%) |  |
| 0.5 | 16.67 ± 0.58 |
| 1.0 | 23.00 ± 0.00 |
| 2.0 | 23.00 ± 0.00 |
| 3.0 | 23.00 ± 0.00 |
| 4.0 | 23.33 ± 0.58 |
| 5.0 | 23.33 ± 0.58 |

**Notes.**
*The data obtained from triplicates and represented as mean ± SD.

*Streptomyces* sp. KB1 (TISTR 2304) into 250, 500, and 1,000 ml baffled flasks containing 50, 100, and 200 ml of LB/2 broth medium, respectively, and incubated them at 25, 30, and 35 °C, 200 rpm in a shaking incubator for 14 days. The effects of incubation period and container volume on the production of BCs of *Streptomyces* sp. KB1 (TISTR 2304) was estimated using the basis of diameter of inhibition zone at temperatures of 25, 30, and 35 °C (Table 6). The obtained results showed that *Streptomyces* sp. KB1 (TISTR 2304) could not produce BCs when it was cultured at a temperature of 35 °C. However, *Streptomyces* sp. KB1 (TISTR 2304) could produce BCs in a growth-phase dependent manner at temperatures of 25 and 30 °C. The results also revealed that the production of BCs of *Streptomyces* sp. KB1 (TISTR 2304) initially started on the 2nd to 3rd day of cultivation, whereas the maximum

anti-*S. aureus* TISTR 517 activity was recorded on the 2nd to 5th day of cultivation and depended on incubation temperature and ratio of medium volume per container volume. The incubation period for production of BCs seemed to be different across *Streptomyces* species (*Shin, Lee & Lee, 2010*). Generally, it was observed that *Streptomyces* species showed a progressive increase of biomass during the first 3 to 5 days of incubation. BC production usually started on the 5th to 7th day and maximum activity was often recorded on the 7th to 14th day (*Sultan et al., 2002*; *Oskay, 2011*; *Rakesh et al., 2014*; *Bundale et al., 2015*). We also found that two phases were observed during the propagation of BC producers. The first phase (trophophase) was characterized by rapid growth (biomass production) and the second phase (idiophase) was characterized by a slow growth and maximal productivity of BCs (*Bundale et al., 2015*). In this study, day 2 to 5 was the suitable incubation period for BC production of *Streptomyces* sp. KB1 (TISTR 2304). This may be due to the decrease of formula of LB medium that stimulated the onset of the second phase of cells. When considering the effect of incubation temperature, it was found that *Streptomyces* sp. KB1 (TISTR 2304) could produce BCs at incubation temperatures of 25 °C and 30 °C, but could not produce at incubation temperatures of 35 °C. In addition, we also found that the anti-*S.aureus* TISTR 517 activity of BCs obtained from the cultivating *Streptomyces* sp. KB1 (TISTR 2304) at an incubation temperature of 30 °C was significantly higher than than at 25 °C. The obtained results revealed that the increase of the incubation temperature from 25 °C to 30 °C and 30 °C to 35 °C increased and completely suppressed the production of BCs of *Streptomyces* sp. KB1 (TISTR 2304). This finding indicated that *Streptomyces* sp. KB1 (TISTR 2304) showed a narrow range of incubation temperatures for BC production and the most suitable incubation temperature for the maximum production of BCs was 30 °C. This finding coincided with research by *Raghavarao et al. (2015)* who reported that *Streptomyces coelicoflavus* BC01 could produce BCs at incubation temperatures between 15 °C to 45 °C, but the optimal temperature was 30 °C. In addition, this finding was in accordance with the results of *Bundale et al. (2015)* who reported that the optimal temperature for the production of BCs for the three *Streptomyces* species in their study was 30 °C. They suggested that there is a wide range of temperature supporting good growth but the temperature range adequate for good production of BCs is narrow, about 5 −10 °C from the optimal temperature for BC production (*Bundale et al., 2015*). When at an incubation temperature of 30 °C, each 1% inoculum of *Streptomyces* sp. KB1 (TISTR 2304) was cultured in three containers of baffled flask (250, 500 and 1,000 ml) which had 50, 100, and 200 ml of LB/2 medium, respectively. All had a medium volume container volume ratio of 1:5, which is a standard ratio for the cultivation of obligate aerobe bacteria. After *Streptomyces* sp. KB1 (TISTR 2304) was cultured in a shaking incubator at 200 rpm, the clear culture broth was investigated daily for anti-*S. aureus* TISTR 517 activity. We observed anti-*S. aureus* TISTR 517 activity starting the 4th day when it was cultured in a 250 ml baffled flask containing 50 ml of LB/2 medium. Additionally, clear culture broth obtained by cultivating *Streptomyces* sp. KB1 in 500 and 1,000 ml baffled flasks containing 100 and 200 ml of LB/2 medium, respectively, initially showed anti-*S. aureus* TISTR 517 activity since the 2nd day of cultivation. However, the significantly highest amount of anti-*S. aureus* TISTR 517 activity was observed when *Streptomyces* sp. KB1 (TISTR 2304) was cultured in a

1,000 ml baffled flask containing 200 ml of LB/2 medium starting the 4th day of cultivation. These results indicated that container volume affected the production of *Streptomyces* sp. KB1 BCs. Several other studies reported that most BC-producing microorganisms are aerobic microorganisms. *Chen & Wilde (1991)* found that oxygen has a significant impact on biosynthetic enzymes involved in BC production. Moreover, it was also found that the growth rate and tyrosine production of *Streptomyces fradiae* was significantly lower at low oxygen concentration when compared with high oxygen concentration (*Chen & Wilde, 1991*). It was suggested that oxygen is an important factor affecting growth and BC production of *Streptomyces* species (*Martins et al., 2004*), which could explain why a large container volume better supports dissolved oxygen for *Streptomyces* sp. KB1 (TISTR 2304) compared to a small container volume. This caused the production of BCs of *Streptomyces* sp. KB1 (TISTR 2304) in a 1,000 ml baffled flask containing 200 ml of LB/2 medium to be higher than the 500 and 250 ml baffled flasks containing 100 and 50 ml of LB/2 medium, respectively. Because the dissolved oxygen influenced BC production of *Streptomyces* sp. KB1 (TISTR 2304), agitation was optimized in the next step by cultivating *Streptomyces* sp. KB1 in a 1,000 ml baffled flask containing 200 ml of LB/2 medium at 30 °C for 4 days in a shaking incubator at different agitation rates.

### Effect of agitation on BC production

The effect of agitation on BC production was determined by cultivating 1% inoculum of *Streptomyces* sp. KB1 (TISTR 2304) in a 1,000 ml baffled flask containing 200 ml of LB/2 medium and incubating it in a shaking incubator at different agitation rates (0, 100, 150, 200, and 250 rpm), 30 °C for 4 days. The diameter of the zone of inhibition was used to estimate the BC production of *Streptomyces* sp. KB1, presented in Table 7. It was found that the agitation at 200 rpm showed the maximum diameter of inhibition zone, indicating that this was the optimal agitation rate for the production of BCs of *Streptomyces* sp. KB1 (TISTR 2304). Agitation affected the mixing of nutrients and amount of dissolved oxygen in the medium, which directly influenced BC production (*Raghavarao et al., 2015*). This finding was in accordance with research from *Augustine, Bhavsar & Kapadnis (2005)* who reported that *Streptomyces rochei* AK39 could produce the maximum amount of cephalosporin C at an agitation rate of 200 rpm. Moreover, this finding also coincided with research from *Raghavarao et al. (2015)* who reported that the production of antibacterial metabolites by *Streptomyces coelicoflavus* BC01 decreased when agitation speed increased (*Raghavarao et al., 2015*). The decrease of BC production of *Streptomyces* species at high agitation speeds indicated that it was affected by the death or damage of cells by shearing forces caused from agitation at high speed (*Lee et al., 2014*).

### Effect of initial pH of medium on BC production

The effect of initial medium pH on BC production of *Streptomyces* sp. KB1 (TISTR 2304) was estimated using the basis of diameter of inhibition zone (Table 7). It was found that *Streptomyces* sp. KB1 (TISTR 2304) could produce BCs when the initial pH of LB/2 medium was in the range 5–9. However, the maximum number of BCs was obtained when the LB/2 medium's initial pH was adjusted to 7. The pH of the medium greatly influenced the production of BCs of *Streptomyces* species by affecting the activity of cellular

**Table 6** **The effect of different incubation periods (d) and medium volume per container volume (ml) on productivity of BCs by *Streptomyces* sp. KB1 (TISTR 2304) against *S. aureus* TISTR 517 under incubation temperatures of 25 °C, 30 °C, and 35 °C.** Each data point indicates the average antimicrobial activity across various conditions. Antimicrobial activity after culturing at different temperatures.

| Medium volume/container volume (ml) | Mean diameter of inhibition zone (mm) at different incubation period (d)* | | | | | | | | | | | | | |
|---|---|---|---|---|---|---|---|---|---|---|---|---|---|---|
| | 1 | 2 | 3 | 4 | 5 | 6 | 7 | 8 | 9 | 10 | 11 | 12 | 13 | 14 |
| 25 °C | | | | | | | | | | | | | | |
| 50/250 of baffled flask | 0.00 ± 0.00 | 0.00 ± 0.00 | 16.33 ± 0.58 | 17.33 ± 0.58 | 17.67 ± 0.58 | 17.67 ± 0.58 | 17.33 ± 0.58 | 17.33 ± 0.58 | 17.33 ± 0.58 | 17.00 ± 0.00 | 16.33 ± 0.58 | 16.33 ± 0.58 | 16.00 ± 0.00 | 15.67 ± 0.58 |
| 100/500 of baffled flask | 0.00 ± 0.00 | 0.00 ± 0.00 | 15.33 ± 0.58 | 15.33 ± 0.58 | 15.67 ± 0.58 | 15.67 ± 0.58 | 15.67 ± 0.58 | 15.00 ± 0.00 | 15.00 ± 0.00 | 14.67 ± 0.58 | 14.67 ± 0.58 | 14.67 ± 0.58 | 14.00 ± 0.00 | 14.00 ± 0.00 |
| 200/1,000 of baffled flask | 0.00 ± 0.00 | 0.00 ± 0.00 | 11.33 ± 0.58 | 11.67 ± 0.58 | 11.67 ± 0.58 | 12.33 ± 0.58 | 12.00 ± 0.00 | 12.00 ± 0.00 | 12.00 ± 0.00 | 12.00 ± 0.00 | 11.67 ± 0.58 | 11.67 ± 0.58 | 11.67 ± 0.58 | 11.00 ± 0.00 |
| 30 °C | | | | | | | | | | | | | | |
| 50/250 of baffled flask | 0.00 ± 0.00 | 0.00 ± 0.00 | 0.00 ± 0.00 | 15.33 ± 0.58 | 16.67 ± 0.58 | 16.67 ± 0.58 | 17.33 ± 0.58 | 17.67 ± 0.58 | 17.33 ± 0.58 | 17.67 ± 0.58 | 17.67 ± 0.58 | 16.67 ± 0.58 | 16.33 ± 0.58 | 16.00 ± 0.00 |
| 100/500 of baffled flask | 0.00 ± 0.00 | 18.00 ± 0.00 | 18.67 ± 0.58 | 18.67 ± 0.58 | 18.67 ± 0.58 | 18.67 ± 0.58 | 19.00 ± 0.00 | 19.00 ± 0.00 | 19.00 ± 0.00 | 19.00 ± 0.00 | 18.67 ± 0.58 | 18.67 ± 0.58 | 18.33 ± 0.58 | 18.33 ± 0.58 |
| 200/1,000 of baffled flask | 0.00 ± 0.00 | 21.00 ± 1.00 | 22.00 ± 1.00 | 23.67 ± 0.58 | 23.67 ± 0.58 | 23.67 ± 0.58 | 23.33 ± 0.58 | 23.67 ± 0.58 | 23.67 ± 0.58 | 23.33 ± 0.58 | 23.33 ± 0.58 | 23.33 ± 0.58 | 23.33 ± 0.58 | 22.33 ± 0.58 |
| 35 °C | | | | | | | | | | | | | | |
| 50/250 of baffled flask | 0.00 ± 0.00 | 0.00 ± 0.00 | 0.00 ± 0.00 | 0.00 ± 0.00 | 0.00 ± 0.00 | 0.00 ± 0.00 | 0.00 ± 0.00 | 0.00 ± 0.00 | 0.00 ± 0.00 | 0.00 ± 0.00 | 0.00 ± 0.00 | 0.00 ± 0.00 | 0.00 ± 0.00 | 0.00 ± 0.00 |
| 100/500 of baffled flask | 0.00 ± 0.00 | 0.00 ± 0.00 | 0.00 ± 0.00 | 0.00 ± 0.00 | 0.00 ± 0.00 | 0.00 ± 0.00 | 0.00 ± 0.00 | 0.00 ± 0.00 | 0.00 ± 0.00 | 0.00 ± 0.00 | 0.00 ± 0.00 | 0.00 ± 0.00 | 0.00 ± 0.00 | 0.00 ± 0.00 |
| 200/1,000 of baffled flask | 0.00 ± 0.00 | 0.00 ± 0.00 | 0.00 ± 0.00 | 0.00 ± 0.00 | 0.00 ± 0.00 | 0.00 ± 0.00 | 0.00 ± 0.00 | 0.00 ± 0.00 | 0.00 ± 0.00 | 0.00 ± 0.00 | 0.00 ± 0.00 | 0.00 ± 0.00 | 0.00 ± 0.00 | 0.00 ± 0.00 |

**Table 7  Effect of agitation and pH on production of antimicrobial compounds.** KB1 grew well under agitation at a neutral pH.

|  | Mean diameter of inhibition zone (mm)* |
| --- | --- |
| **Agitation (rpm)** |  |
| 0 | 0.00 ± 0.00 |
| 100 | 17.33 ± 0.58 |
| 150 | 21.00 ± 0.00 |
| 200 | 23.33 ± 0.58 |
| 250 | 22.67 ± 0.58 |
| **Initial pH of LB/2 medium** |  |
| 5 | 11.67 ± 0.58 |
| 6 | 19.33 ± 0.58 |
| 7 | 23.00 ± 0.00 |
| 8 | 21.33 ± 0.58 |
| 9 | 18.00 ± 0.00 |

**Notes.**
*The data obtained from triplicates and represented as mean ± SD.

enzymes correlated with the biosynthesis of BCs (*Bundale et al., 2015*). Several studies reported that the optimal initial medium pH for BC production of the *Streptomyces* species depended on their optimal growth in a neutral environment (*Sultan et al., 2002*; *Oskay, 2011*; *Reddy, Ramakrishna & Rajagopal, 2011*). Therefore, we found that most BCs were optimally produced at a pH level close to 7.

### Effect of carbon sources on BC production

The effect of different carbon sources on the BC production of *Streptomyces* sp. KB1 was estimated by using the basis of diameter of inhibition zone (Table 8). The obtained results demonstrated that the BC production of *Streptomyces* sp. KB1 (TISTR 2304) decreased when the LB/2 medium was supplemented with different carbon sources. Moreover, we found that *Streptomyces* sp. KB1 could not produce BCs when the LB/2 medium was supplemented with glucose as its carbon source. BC biosynthesis strongly correlated with components and their concentrations in culture media (*Rafieenia, 2013*). A carbon source usually constitutes the major part of a culture medium because it can readily serve as growth substrates. Several studies have shown the effect of different carbon sources on the production of BCs of *Streptomyces* species (*Jonsbu, McIntyre & Nielsen, 2002*; *Sharon, Daniel & Shenbagarathai, 2014*; *Raghavarao et al., 2015*). It was found that simply metabolizable carbon sources generally repressed secondary metabolism or the biosynthesis of BCs (*Escalante et al., 1982*). Several researchers reported that glucose and other monosaccharides strongly decreased the production of BCs such as oleandomycin, avilamycin, nystatin, spiramycin and neomycin (*Rafieenia, 2013*). It has been indicated that this is caused by carbon catabolite repression (*Escalante et al., 1999*; *Bruckner & Titgemeyer, 2002*). Carbon catabolite repression is the synthesis of BC biosynthesis-dependent enzymes being repressed by the presence of a catabolite, usually generated from a rapidly metabolizable exogenous carbon source such as glucose. Catabolite repression was first shown to be initiated by glucose and is therefore sometimes referred to as the glucose

effect (*Martin & Demain, 1980*; *Escalante et al., 1999*). The mechanism of carbon catabolic repression for regulating BC biosynthesis was described by *Saier (1998)*, who suggested that the biosynthesis of BCs was related to phosphoenolpyruvate sugar phosphotransferase, and controlling the concentration of carbon sources could regulate BC biosynthesis. It was found that different carbon sources have widely affected the production of BCs. For example, *Selvin, Shanmughapriya & Gandhimathi (2009)* reported that the maximal production of antimicrobial compounds of *Streptomyces* sp. strain US80 and *Streptomyces* sp. strain TN97 was obtained in a medium supplemented with glucose and glycerol or fructose, respectively. *Kathiresan, Balagurunathan & Selvam (2005)* reported that among the carbon sources, viz. glycerol, glucose, lactose, and maltose, glucose was the best carbon source for BC production. *Himabindu & Jetty (2006)* reported that starch was the best carbon source for growth rate and gentamicin production, and the production was lowered when glucose was used as carbon source instead of starch. Among the carbon sources, it is well known that glucose is the best for supporting cell growth while BC production is not absolute (*Huck et al., 1991*). Therefore, some have studies reported that BC production was not apparent when the culture media was supplemented with different carbon sources such as glucose, lactose, galactose, raffinose, and maltose as the sole carbon source at a concentration of 1% (w/v) (*Martin & Demain, 1980*). In this study, we found that *Streptomyces* sp. KB1 could produce the maximum number of BCs in LB/2 medium when it was not supplemented with a carbon source. The obtained results also found that the production of BCs of *Streptomyces* sp. KB1 consequently decreased when the LB/2 medium was supplemented with polysaccharides (glycerol and starch) and disaccharides (maltose and sucrose), and did not occur when the LB/2 medium was supplemented with monosaccharide (glucose). This finding was discussed by *Rafieenia (2013)* who explained that a slow growth rate affected BC production of *Streptomyces* species. Since a large molecule could support the slow growth rate desirable for BC production, polysaccharides support the production of BC production of *Streptomyces* sp. KB1 better than disaccharides and monosaccharides (*Rafieenia, 2013*). However, the maximum number of BCs were obtained by cultivating *Streptomyces* sp. KB1 in a LB/2 medium that was not supplemented with carbon sources. Therefore, the further optimization process was carried out by cultivating *Streptomyces* sp. KB1 in LB/2 medium with no supplemental carbon source.

### Effect of nitrogen sources on BC production

The effect of different organic and inorganic nitrogen sources on the BC production of *Streptomyces* sp. KB1 (TISTR 2304) was estimated using the diameter of the zone of inhibition presented in Table 8. In this study, each nitrogen source was supplemented at a concentration of 0.3% (w/v) because many previous studies have shown that the biosynthesis of BCs strongly related to the nature, type, and concentration of nitrogen source in the culture of *Streptomyces* (*Rafieenia, 2013*). Additionally. the presence of excess nitrogen sources in a culture medium decreased BC production (*Aharonowitz, 1980*; *Martin & Demain, 1980*; *Doull & Vining, 1990*; *Spizek & Tichy, 1995*; *Ripa et al., 2009*). The obtained results demonstrated that *Streptomyces* sp. KB1's (TISTR 2304) ability to produce BCs was not significantly different when it was cultivated in LB/2 medium and

**Table 8 Effect of carbon sources, nitrogen sources, salt concentrations, or trace elements on BC production of _Streptomyces_ sp. KB1 TISTR 2304.** Each data point indicates the average antimicrobial activity across various conditions.

|  | Mean diameter of inhibition zone (mm)* |
|---|---|
| **Carbon sources (1%)** | |
| Glucose | $0.00 \pm 0.00$ |
| Maltose | $15.67 \pm 0.58$ |
| Sucrose | $14.00 \pm 1.00$ |
| Glycerol | $19.33 \pm 0.58$ |
| Starch | $20.67 \pm 0.58$ |
| No supplement | $23.33 \pm 0.58$ |
| **Nitrogen sources (0.3%)** | |
| Casein | $22.67 \pm 0.58$ |
| Peptone | $22.67 \pm 0.58$ |
| Beef extract | $21.33 \pm 0.58$ |
| Malt extract | $22.67 \pm 0.58$ |
| Ammonium sulphate | $20.67 \pm 0.58$ |
| Urea | $19.67 \pm 0.58$ |
| No supplement | $23.00 \pm 0.00$ |
| **Salt concentration (%)** | |
| 0.0 | $23.00 \pm 0.00$ |
| 0.1 | $23.00 \pm 1.00$ |
| 0.5 | $22.67 \pm 0.58$ |
| 1.0 | $21.00 \pm 1.00$ |
| 1.5 | $18.33 \pm 0.58$ |
| 2.0 | $15.00 \pm 1.00$ |
| 2.5 | $11.67 \pm 0.58$ |
| 3.0 | $0.00 \pm 0.00$ |
| **0.1% of trace elements** | |
| $FeSO_4$ | $0.00 \pm 0.00$ |
| $MgSO_4$ | $23.00 \pm 1.00$ |
| $CaCO_3$ | $23.00 \pm 0.00$ |
| $K_2HPO_4$ | $0.00 \pm 0.00$ |
| $KH_2PO_4$ | $19.33 \pm 0.58$ |
| KCl | $0.00 \pm 0.00$ |
| $KNO_3$ | $0.00 \pm 0.00$ |
| No supplement | $23.00 \pm 0.00$ |

Notes.

*The data obtained from triplicates and represented as mean $\pm$ SD.

LB/2 medium supplemented with casein, peptone, and malt extract. The finding indicated that this organic nitrogen source did not affect the production of BCs of _Streptomyces_ sp. KB1 (TISTR 2304). However, BC production of _Streptomyces_ sp. KB1 (TISTR 2304) significantly decreased when it was cultivated in LB/2 medium supplemented with beef extract, ammonium sulphate, and urea. This phenomenon could be due to the fact that

beef extract might decompose and metabolize faster than other organic nitrogen sources because quickly decomposable and metabolizable nitrogen sources usually decrease the production of BCs. Meanwhile, the use of ammonium sulphate and urea, which are inorganic nitrogen sources, create high ammonium concentrations in culture medium and suppress BC production in many microorganisms including *Streptomyces* sp. KB1 (TISTR 2304) (*Rafieenia, 2013*; *Raghavarao et al., 2015*).

### Effect of salt concentration on BC production

The effect of salt concentration on the BC production of *Streptomyces* sp. KB1 was determined using the basis of diameter of zone of inhibition presented in Table 8. The obtained results indicated that NaCl concentration greatly influenced the production of BCs and *Streptomyces* sp. KB1 could produce BCs when it was cultivated in a LB/2 medium supplemented with NaCl at a concentration lower than 2.5% (w/v). Additionally, the maximum production of BCs was observed at NaCl concentrations of 0.0, 0.1, and 0.5% (w/v) and gradually decreased when the NaCl concentration increased. Salt concentration has a profound effect on the production of BCs of many microorganisms due to its effect on the osmotic pressure in the culture medium, which continuously correlates with the production of BCs (*Pelczar, Chan & Krieg, 1993*). When considering other research, we found that the suitable NaCl concentration for the production of BCs was in the range of 1–2% (w/v), but depended on the origin source of microorganisms (*Ripa et al., 2009*; *Reddy, Ramakrishna & Rajagopal, 2011*; *Sharon, Daniel & Shenbagarathai, 2014*): (*Raghavarao et al., 2015*). This finding revealed that the suitable NaCl concentration for BC production of *Streptomyces* sp. KB1 (TISTR 2304) was in the range of 0 - 0.5% (w/v), which could explain why the LB/2 medium had contained the NaCl at a concentration of 5 g/L. Moreover, we found that the NaCl requirement of *Streptomyces* sp. KB1 (TISTR 2304) was lower than that of the other marine *Streptomyces* because the source of *Streptomyces* sp. KB1 (TISTR 2304) was an air sample.

### Effect of trace elements on BC production

The effect of trace elements on the BC production of *Streptomyces* sp. KB1 (TISTR 2304) was estimated using the basis of diameter of zone of inhibition presented in Table 8. The results demonstrated that $MgSO_4$ and $CaCO_3$ did not influence the production of BCs of *Streptomyces* sp. KB1 (TISTR 2304) because the diameter of inhibition zone was not significantly different when compared with the control group (LB/2 medium with no supplement of trace elements). However, the BC production of *Streptomyces* sp. KB1 decreased when it was cultivated in LB/2 medium supplement with $KH_2PO_4$. Surprisingly, *Streptomyces* sp. KB1 could not produce BCs when the LB/2 medium was supplemented with $FeSO_4$, $K_2HPO_4$, KCl, and $KNO_3$. *Streptomyces* species require some specific trace elements for their growth and antimicrobial metabolite production (*Basak & Majumdar, 1975*; *Narayana & Vijayalakshmi, 2008*; *Bundale et al., 2015*). This requirement varies with the type of organism as well as the nature of basal medium used. For example, zinc (Zn) favorably supported the growth of *Streptomyces griseus* but inhibited the production of BCs. While Zn is essential for the BC production of *Streptomyces fradiae*, it is not affected by the growth of this organism (*Chesters & Rolinson, 1951*; *Majumdar & Majumdar, 1965*).

Moreover, it was found that iron (Fe) and $K_2HPO_4$ at concentrations of 0.25 µg/ml and 1.0 mg/ml, respectively, were essential for the maximum BC production of *Streptomyces kanamyceticus*, whereas the production of BCs of this organism continuously decreased when Fe and $K_2HPO_4$ concentrations significantly increased. From this same research, we also found that KCl did not affect the production of BCs of *Streptomyces kanamyceticus*, but only negatively affected it when the concentration was higher than 5 mg/ml (*Basak & Majumdar, 1975*). Therefore, the discussion above could explain why the production of BCs of *Streptomyces* sp. KB1 (TISTR 2304) might have been inhibited by excess $FeSO_4$, $K_2HPO_4$, KCl, and $KNO_3$ in this study. Meanwhile, the BC production of *Streptomyces* sp. KB1 was not significantly different between the LB/2 medium supplemented with $MgSO_4$ and $CaCO_3$, and the control group (LB/2 medium not supplemented with $MgSO_4$ and $CaCO_3$). This means that $MgSO_4$ and $CaCO_3$ at a concentration of 0.1% had no effect on the production of BCs of *Streptomyces* sp. KB1. To study the effect of trace elements on BC production, a synthetic medium is usually selected. However, in this study, we used complex media (LB/2 medium) that might have contained considerable amounts of some essential trace elements as contaminants. Therefore, it has been revealed that *Streptomyces* sp. KB1 could produce the maximum number of BCs in LB/2 medium with no supplement of trace elements.

## CONCLUSION

*Streptomyces* sp. KB1 (TISTR 2304) is a gram-positive and long filamentous bacteria that forms straight to flexuous (rectiflexibile) chains of globose-shaped and smooth-surfaced spores that grow under aerobic conditions only at a temperature range of 25–37 °C and initial pH range of 5–10 in the presence of sodium chloride <4% (w/v). It is considered an obligate aerobe, mesophilic, neutralophilic, and moderate halophilic bacteria that can well grow on PYEI, LB, and LB/2 medium. *Streptomyces* sp. KB1 (TISTR 2304) can utilize fructose, mannose, glucose, and lactose as its carbon source along with acid production and also showed positive casein hydrolysis, gelatin liquefaction, nitrate reduction, urease, and catalase production results. In addition, *Streptomyces* sp. KB1 (TISTR 2304) could produce the maximum number of BCs when 1% of its starter was cultivated in a 1,000 ml baffled flask containing 200 ml of LB/2 broth with an adjusted initial pH of 7 and no supplemental carbon source, nitrogen source, NaCl, or trace element at 30 °C at 200 rpm in a shaking incubator for 4 days.

### Funding

This research was supported by Plant Genetic Conservation Project Under the Royal Initiation of Her Royal Highness Princess Maha Chakri Sirindhorn (RSPG): WUBG-019/2564, WUBG-008/2565, RSPG 21-2566. The funders had no role in study design, data collection and analysis, decision to publish, or preparation of the manuscript.

## Grant Disclosures

The following grant information was disclosed by the authors:
Plant Genetic Conservation Project Under the Royal Initiation of Her Royal Highness Princess Maha Chakri Sirindhorn (RSPG): WUBG-019/2564, WUBG-008/2565, RSPG 21-2566.

## Competing Interests

The authors declare there are no competing interests.

## Author Contributions

- Monthon Lertcanawanichakul conceived and designed the experiments, performed the experiments, analyzed the data, prepared figures and/or tables, authored or reviewed drafts of the article, and approved the final draft.
- Tuanhawanti Sahabuddeen performed the experiments, analyzed the data, prepared figures and/or tables, and approved the final draft.

## Data Availability

The raw data are available in the Supplemental Files.

## Supplemental Information

Supplemental information for this article can be found online at http://dx.doi.org/10.7717/peerj.14909#supplemental-information.

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

## FURTHER READING

**Procopio REL, Silva IR, Martins MK, Azevedo JL, Araujo JM. 2012.** Antibiotics produced by *Streptomyces*. *The Brazilian Journal of Infectious Diseases* **16(5)**:466–471 DOI 10.1016/j.bjid.2012.08.014.

**Smanski MJ, Peterson RM, Rajski SR, Shen B. 2009.** Engineered *Strep-tomyces platensis* strains that overproduce antibiotics platensimycin and platencin. *Journal of Bioactive Agent and Chemotherapy* **53(4)**:1299–1304 DOI 10.1128/AAC.01358-08.