# Peer review of "Characterization of Streptomyces sp. KB1 and its cultural optimization for bioactive compounds production"

_PeerJ, doi:10.7717/peerj.14909_

## Round 0.1 · original submission · Major Revisions

Check the reviewers' comments: English used in the manuscript needs to be thoroughly revised and improved and see if all the reviewers' comments can be addressed. The novelty and significance of the work need to be highlighted.

Reviewer 1 ·

Basic reporting

English used in the manuscript needs to be thoroughly revised and improved.
The sentences are not clear due to poor English.

Do not use first person in the manuscript. E.g. Line no. 189. Correct it

Cite the following article in the introduction
Sharma, P., Thakur, D. Antimicrobial biosynthetic potential and diversity of culturable soil actinobacteria from forest ecosystems of Northeast India. Sci Rep 10, 4104 (2020)

Experimental design

What is the positive and negative control used for studying antimicrobial activity?

Line 58: Please write the correct order name.

Please don’t write ‘gene’ after 16s rDNA. Correct it throughout the manuscript. E.g. line no. 60, 102, 109 etc.

Line no. 59 – 61 and Line no. 101 - 104 not necessary. It has no relevance to the paper.

Validity of the findings

What is the novelty and signifcance of the work?

Figure 1: Include Scale bar, magnification

Figure 2: Write the inoculum% used to study colony morphology of Streptomyces in different types of media

Line no. 334: Change the caption

Write what is S and R in Table 2

Reviewer 2 ·

Basic reporting

The work is quite a basic study only; but is better articulated.

Experimental design

The work requires quantification & identification of bioactive compounds for completion of this work.
Kindly look into works done by zothanpuia et al 2018; where MIC concentration are quantified. I would recommend using methodologies similar to that in future studies too.

Validity of the findings

I would recommend including a few images of the zone of inhibitions from each categories. Because by simply stating zone existence in tables is not enough to prove their viability.

Additional comments

The phrase "Bioactive compounds" is mentioned quite vociferously; without significant identification or quantification of it. But hope the current work will be the stepping stone to this identification. Hereby requesting image evidences to substantiate the zone of inhibitions under various conditions enlisted in the methodology.

---

## Round 0.2 · Minor Revisions

Kindly note the comments of reviewer 1 and if can be addressed can be considered for publication.

Reviewer 1 ·

Basic reporting

As mentioned earlier, English needs to be improved ‘all throughout’ the manuscript, including the sentences that are highlighted and marked red by the authors.

Experimental design

Okay

Validity of the findings

Figure 3: Remove the black dot from the well of culture broth.

Additional comments

None

Reviewer 2 ·

Basic reporting

No comment

Experimental design

No comment

Validity of the findings

No comment

---

## Round 0.3 · accepted · Accept

Authors have addressed all of the reviewers' comments and the manuscript is ready for publication.

Reviewer 1 ·

Basic reporting

English is corrected

Experimental design

Looks fine

Validity of the findings

Looks fine

Additional comments

No